# Diagnostic Study of Mandibular Cortical Index Classification Using Dental Cone-Beam Computed Tomography Findings: A Preliminary Cross-Sectional Study

**DOI:** 10.3390/reports6040048

**Published:** 2023-10-11

**Authors:** Keisuke Seki, Tona Yoshino, Shouhei Ogisawa, Yushi Arai, Morio Tonogi, Toshimitsu Iinuma

**Affiliations:** 1Nihon University School of Dentistry, Mishima Dental Center, 1-9-18, Bunkyo-cho, Mishima-shi 411-8588, Japan; yoshino.tona@nihon-u.ac.jp (T.Y.); tonogi.morio@nihon-u.ac.jp (M.T.); iinuma.toshimitsu@nihon-u.ac.jp (T.I.); 2Department of Comprehensive Dentistry and Clinical Education, Nihon University School of Dentistry, 1-8-13, Kanda-Surugadai, Chiyoda-ku, Tokyo 101-8310, Japan; 3Division of Dental Education, Dental Research Center, Nihon University School of Dentistry, 1-8-13, Kanda-Surugadai, Chiyoda-ku, Tokyo 101-8310, Japan; 4Department of Oral and Maxillofacial Surgery I, Nihon University School of Dentistry, 1-8-13, Kanda-Surugadai, Chiyoda-ku, Tokyo 101-8310, Japan; ogisawa.shouhei@nihon-u.ac.jp; 5Nihon University School of Dentistry Dental Hospital, 1-8-13, Kanda-Surugadai, Chiyoda-ku, Tokyo 101-8310, Japan; arai.yuushi.resi23069@nihon-u.ac.jp; 6Department of Complete Denture Prosthodontics, Nihon University School of Dentistry, 1-8-13, Kanda-Surugadai, Chiyoda-ku, Tokyo 101-8310, Japan

**Keywords:** cone-beam computed tomography, cross-sectional studies, mandible, osteoporosis, panoramic, porosity, radiography

## Abstract

The prevalence of osteoporosis is high, reportedly affecting 200 million people worldwide. A major problem associated with osteoporosis is that femoral fractures cause a decline in general function and loss of independence, greatly reducing patients’ quality of life. Notably, osteoporosis is an asymptomatic chronic metabolic disease, and its detection is thus often delayed. Interestingly, mandibular cortical index (MCI) classification using dental panoramic radiography is reportedly useful for early detection of osteoporosis. However, this visual classification method can lead to differences in diagnoses among surgeons. The aim of this preliminary study was to analyze cone-beam computed tomography (CBCT) data and examine an objective MCI classification using the findings obtained. MCI classification (classified as C1, C2, or C3) was performed by three examiners on 70 women (91 sites) aged ≥20 years. The mandibular cortical width of all sites was measured using CBCT images. The results showed that the mandibular cortical width was not particularly correlated with age or number of present teeth, and no significant quantitative differences were found between C1 and C2. However, coronal CBCT images of C2 revealed multiple characteristic trabecular bone structures. These structures may be an important finding affecting the classification of two-dimensional dental panoramic radiography images.

## 1. Introduction

Osteoporosis is a skeletal disorder characterized by compromised bone strength, predisposing an individual to an increased risk of fracture. Bone strength reflects the integration of two main features: bone quantity and bone quality [1]. The pathogenesis of osteoporosis involves an alteration in the balance of normal bone remodeling, with bone resorption exceeding bone formation; this results in reduced bone density and strength [2]. Secondary osteoporosis (related to endocrine factors, nutrition, medicines, immobility, rheumatism, and congenital causes) presents as low bone mass and affects a larger proportion of patients than primary osteoporosis [3]. The prevalence of osteoporosis is high, reportedly affecting 200 million people worldwide [4,5]. A major problem associated with osteoporosis is that femoral fractures cause a decline in general function and loss of independence, greatly reducing patients’ quality of life. Bisphosphonates and denosumab, two types of osteoporosis drugs extensively used worldwide, have great advantages in the prevention of bone fractures; however, severe osteonecrosis of the jaw has been reported in long-term users [6,7]. This problem is of great concern in dentistry. Notably, osteoporosis is an asymptomatic chronic metabolic disease, and its detection is thus often delayed [8]. In recent years, a unique dental approach has attracted attention as a method of identifying undiagnosed osteoporosis. Interestingly, one study involving commonly used dental panoramic radiography (DPR) techniques showed that approximately 90% of patients who exhibited cortical bone rarefaction on evaluation of the shape of the mandibular inferior cortex had osteoporosis [9]. Subsequent studies showed that evaluation of the mandibular cortical index (MCI) by DPR imaging is useful in screening for osteoporosis [10,11,12]. Because the shape of the mandibular inferior cortex reflects the bone density of the lumbar vertebrae and the femur and is associated with bone metabolism markers and fracture risk [13,14], evaluation of the MCI by DPR is a useful screening tool for detecting osteoporosis [11,12,15]. However, although the morphological MCI classification method based on visual inspection of two-dimensional X-ray images is simple, the fact that different examiners have different judgments regarding mandibular width measurements remains a diagnostic problem. The accuracy of 3D reconstructions of the craniomaxillofacial region using cone-beam computed tomography (CBCT) is important for the morphological evaluation of the mandibular cortical bone anatomical structures [16,17,18]. Therefore, the purpose of this preliminary study was to support the DPR image-based MCI classification numerically and to simplify difficult diagnoses through measurement of the mandibular inferior cortical bone width in three dimensions using cone-beam computed tomography (CBCT) images. Furthermore, the ultimate goal was that the imaging findings obtained during dental care would serve as a better diagnostic aid in detecting hidden osteoporosis.

## 2. Materials and Methods

### 2.1. Study Population

This observational study was approved by the Ethics Committee of Nihon University School of Dentistry (Permit No. EP20D006). This study was conducted in accordance with the 2013 revision of the 1975 Declaration of Helsinki and in line with the guidelines for observational/descriptive studies on enhanced reporting of observational studies in epidemiology [19]. All women who visited the Nihon University School of Dentistry Mishima Dental Center from 2015 to 2022 (the clinic closed in March 2022) were included in this study, and the patients’ medical records were collected. Documents describing the collection, purpose of use, and research methods were posted in the clinic, and the information was disclosed and made known to the participants through the clinic website.

### 2.2. Data Sources and Patient Selection

The inclusion criteria for this study were women aged ≥20 years presenting to the authors’ dental center from December 2015 to February 2022, acquisition of CBCT data for dental treatment, and acquisition of DPR data for dental treatment. The exclusion criteria were CT images that did not include the mandibular inferior cortical bone, DPR images in which the morphology of the mandibular inferior cortical bone could not be diagnosed because of ghost images or poor positioning, a history of mandibular body resection or reconstruction or a history of bone-destroying neoplastic lesions, and a history of radiotherapy to the head and neck region.

### 2.3. Image Data Capture

All DPR images and CBCT images used in this study were taken with a ProMax 3D X-ray machine (Planmeca, Helsinki, Finland) in the clinic. The imaging conditions were as follows: DPR images, tube voltage 66 kV, tube current 9 mA, and irradiation time 19 s, CBCT images, voxel size 200 μm × 200 μm × 200 μm, tube voltage 90 kV, tube current 10 mA, and irradiation time 12 s. The diagnostic imaging software used was the Romexis 2D and 3D imaging module (Planmeca). When extracting demographic and medical data, the examiners were blinded to all personal information on the DPR and CBCT images. All data taken were anonymized for ethical reasons and to reduce measurer bias and were managed with a newly assigned serial number. The output image data were evaluated on a diagnostic monitor (EIZO Corporation, Ishikawa, Japan). The number of present teeth was counted by DPR. Because the soft tissue condition could not be determined by DPR images, third molars with the tooth axis perpendicular to the occlusal plane were counted as present teeth. Those with the tooth axis horizontal to the occlusal plane were treated as impacted teeth and were not included in the number of present teeth. Implant placement sites were treated as missing areas and were not included in the number of present teeth. As the steps of the analysis are described below, MCI was evaluated by three examiners, each on the same DPR samples. Separately, one examiner measured the MCW of the CBCT images.

### 2.4. MCI Evaluation

For all DPR images, the mandibular inferior cortical bone morphology was classified into three types according to a report by Klemetti and Kolmakow [20]. Class 1 (C1) has a smooth inner surface of cortical bone, class 2 (C2) has an irregular inner surface of cortical bone with linear resorption, and class 3 (C3) has severe linear resorption and cortical bone rupture over the entire cortical bone (Figure 1A–C). Diffuse opacities, such as sclerosing osteomyelitis that did not correspond to any of the above types, were recorded separately. Two sites on the left and right sides were evaluated and recorded per patient. Therefore, 140 sites were covered in the DPR images. The MCI was assessed twice by dentists (K.S., a periodontist; T.Y., an oral surgeon; and Y.A., a dental trainee) who had been trained in classification, and the second assessment was used. The classification result was replaced with a numerical value, such as 1 for C1, 2 for C2, and 3 for C3, and the averages of the three examiners’ values were rounded to obtain the final MCI (e.g., if the mean value was 1.66, then it was classified as C2). The interindividual reproducibility (kappa coefficient) was examined for the three dentists who made the evaluations in this study. Tests were conducted in combinations of two dentists each (K.S. and T.Y., T.Y. and Y.A., Y.A. and K.S.).

### 2.5. Inter-Rater Reliability

The three examiners performed MCI classification on 30 randomly selected panoramic X-rays and classified them again in a different order 1 week later. Cohen kappa scores were obtained for intraindividual and interindividual reproducibility. Kappa scores of 0.41 to 0.60 indicated fair agreement, 0.61 to 0.80 indicated good agreement, and 0.81 to 0.92 indicated very good agreement [21].

### 2.6. Mandibular Cortical Width Measurements

The thickness of the mandibular inferior cortical bone was measured in all CBCT images. These measurements were performed in two cross sections to obtain the coronal width (CW) and sagittal width (SW) twice on each side in 0.1 mm increments by one examiner (K.S.), using the image analysis software described above (Figure 2A,B). The target area was immediately below the mental foramen (area indicated as red arrow in Figure 2A,B), and the Hounsfield unit (HU) display tool provided with the software was used as an indicator to digitally measure the bone tissues with a value of ≥1000 [22]. The HU value could be easily displayed on the viewer image by manipulating the pointer in any region. Finally, the CW and SW values were averaged to obtain the mandibular cortical width (MCW). When the CBCT imaging site was the whole jaw, two sites were measured (one on the left and one on the right), and when the imaging site was only one side, one site was measured. In some cases, the bone margin immediately below the mental foramen was not included in the imaging area, in which case an adjacent area, such as that immediately below the first molar, was measured.

### 2.7. Statistical Analysis

All statistical analyses were performed using EZR (Saitama Medical Center, Jichi Medical University, Tochigi, Japan), a graphical user interface of R version 4.0.0 (The R Foundation for Statistical Computing, Vienna, Austria) [23]. MCI and MCW obtained by descriptive statistics were matched by site, and the correlation with each variable was determined. The Kolmogorov–Smirnov test was used to assess the normality of the distribution of continuous variables (age, number of teeth, and MCW), and *p* ≥ 0.05 was considered to indicate a normal distribution. Age (*p* = 0.519) and MCW (*p* = 0.787) were normally distributed, but the number of present teeth (*p* < 0.05) showed a non-normal distribution. For this reason, Pearson’s product-moment correlation coefficient was used for age, and Spearman’s rank correlation coefficient was used for the number of present teeth. For MCW, analysis of variance was determined for the three groups (C1–C3). The results of the F-test for the MCWs of C1 and C2 showed homogeneity of variance (*p* = 0.336); therefore, Student’s *t*-test was performed to examine the difference between the two groups. All tests were statistically significant at *p* < 0.05.

## 3. Results

In total, 1163 women were seen during this study period. Of these patients, 105 underwent both DPR and CBCT examinations. Among them, seventy-one patients aged > 20 years met the inclusion criteria, and one patient was excluded because of a previous segmental mandibulectomy (exclusion ratio was 1.4%). Therefore, the final number of patients was 70, and all were Japanese. The total number of measurable mandibular inferior cortical bone sites in the CT images was 91 (44 on the right side and 47 on the left side). This is because some of the CBCT images included both sides in the imaging area, while others included only one side. The patients’ mean age was 44.1 ± 14.9 (median, 44) years, and the mean number of present teeth was 26.8 ± 4.5 (median, 28). The MCI was C1 in fifty-three sites, C2 in thirty-five sites, and C3 in three sites (Table 1). The best kappa coefficient of interindividual reproducibility was 0.575 (T.Y. and Y.A.), indicating fair agreement. The correlation coefficient between MCW and age was −0.116 (*p* = 0.274), and that between MCW and number of present teeth was 0.151 (*p* = 0.154); no correlation between either variable and MCW was found. The analysis of variance was performed on MCW in the three groups (C1–C3); however, only three sites had MCI of C3, resulting in an uncertain test of homogeneity of variance. Thus, only the difference between C1 and C2 was examined, and no significant difference was found between the two groups (*p* = 0.646).

## 4. Discussion

This preliminary single-center cross-sectional study was performed in an attempt to establish an objective MCI classification by digitally measuring CBCT cross-sectional images of the mandibular inferior cortical bones. We hope that the imaging findings obtained during dental care will serve as a better diagnostic aid in detecting hidden osteoporosis. To achieve these objectives, this observational study was conducted as a descriptive epidemiological investigation. Our facility is characterized by a large number of patients who have been referred for advanced oral surgical treatment. Although we did not extract the records of procedures and diseases in this study, most of the cases were requests for extraction of an impacted wisdom tooth. The mean patient age of 44 years and the mean number of present teeth of 26.8 reflected a relatively young patient population. This may be closely related to the fact that only three of the ninety-one sites were classified as having MCI of C3, which is common among older people, and 97% of the sites had MCI of C1 and C2.

In this study, the coronal and sagittal sections of the CBCT images were examined in detail with a digital measuring device on an X-ray viewer. In a previous study, the MCI classification of C3 was difficult to discriminate by DPR images alone [24], and cross-sectional measurement using CBCT images was a more accurate examination method than DPR [25]. In the present study, however, C3 (which had more bone rarefaction) was easier to discriminate visually in the DPR images, whereas C1 and C2 were more difficult to discriminate. Additionally, MCW in CT was not significantly different in the present study. In fact, the inter-rater reproducibility of the classification results by three dentists with different expertise (periodontist, oral surgeon, and dental trainee) was moderately consistent. This suggests that it is almost impossible to visually distinguish the difference in the cortical bone width of the mandible in DPR images. The mean MCW of C3 was 1.2 to 1.3 mm smaller than that of C1 and C2, suggesting that visual identification of C3 is relatively easy.

Among the CBCT images obtained in this study, the coronal section of C2 showed characteristic findings. The cross section of the cortical bone of the mandibular submarginal margin showed a clean structure with a constant width in C1. However, a distinct trabecular structure was observed in C2, crossing the buccolingual side in the bone marrow. The origin of this structure is not known, but we speculate that it may have remained as a random structure after it was left behind by the process of rarefaction and resorption of the trabecular bone. The blurred and overlapping linear findings in the upper part of the mandibular inferior cortical bone observed in the DPR image of C2 were presumed to be a two-dimensional projection of this structure. These results suggest that when classifying the MCI on DPR images, especially when discriminating between C1 and C2, there is no visible difference in the bone width of the inferior mandible, and the presence or absence of discontinuous resorption fossae or overlapping linear structures is helpful for classification. Despite the small sample size, MCW of C3 was 2.0 to 2.3 mm. In the CT cross-sectional view, the trabecular bone area was also markedly coarsened and clearly distinguishable from C1 and C2. The fact that only C3 was easily discriminated even in the two-dimensional panoramic images was considered to support the fact that the structural changes within the mandible were clearly reflected in these images. Although the MCI classification is useful as a simple diagnostic tool for osteoporosis, detailed diagnostic imaging findings may be important and require experience. We believe that understanding the characteristic structures within the mandibular body shown in this study will enable more accurate MCI classification using DPR. Elderly patients with osteoporosis have been reported to have poor medication compliance [26,27], even when using bone resorption inhibitors that may be causative agents of medication-related osteonecrosis of the jaw, and they are often unaware of their own medications. Therefore, we speculate that MCI classification using DPR may be useful not only for identifying patients with undiagnosed osteoporosis, but also for screening patients who are undergoing osteoporosis treatment using drugs associated with the development of medication-related osteonecrosis of the jaw.

The strength of this study is that it will contribute to presenting evidence in an unknown area that links the fields of osteoporosis treatment and dentistry. The main limitation of this study is that category C3, which indicates a particularly high degree of bone rarefaction, was uncommon among the patients. This MCI category is predicted to be more common in elderly patients. Future studies should clarify the diagnostic features in this regard. Because this was a preliminary study, the differences among all the classifications have not yet been clarified. This problem can be solved by increasing the number of cases; therefore, future multicenter studies should be conducted to clarify this issue.

## 5. Conclusions

The digitally measured MCW showed no particular correlation with age or number of present teeth, and no significant difference was found between C1 and C2 of the MCI classification. However, multiple characteristic trabecular bone structures were observed in the CBCT coronal section image of C2, suggesting that these structures may be important findings when discriminating between the two-dimensional DPR images. In the future, dentists treating older people should be aware of these characteristics, which may contribute to reducing the severity of osteoporosis.

## Figures and Tables

**Figure 1 reports-06-00048-f001:**
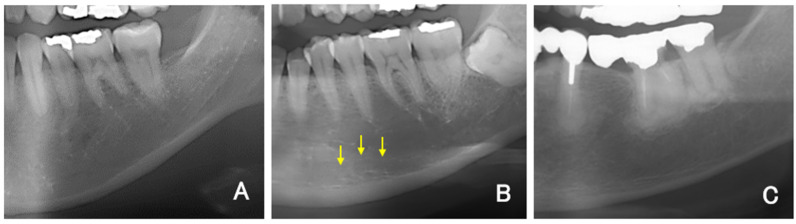
(**A**) Representative example of C1. (**B**) Representative example of C2. Multiple upper marginal lines of the mandibular inferior cortical bone were identified and obscured (arrows). (**C**) Representative example of C3.

**Figure 2 reports-06-00048-f002:**
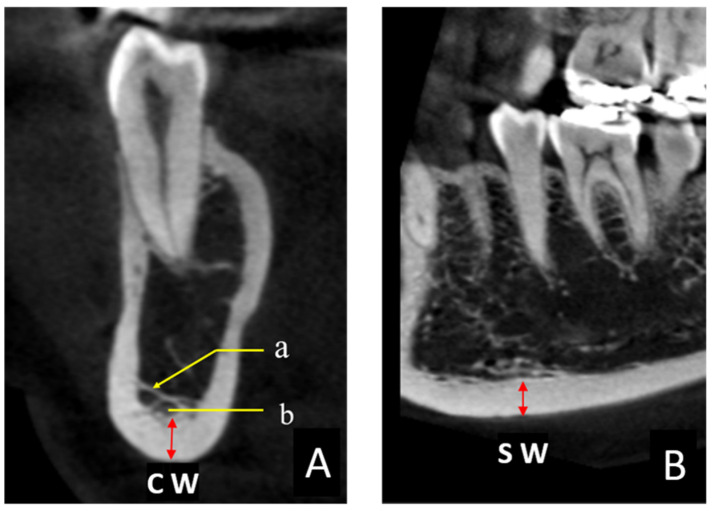
(**A**) CBCT coronal section of a C2 (Figure 1B) case. (a) Trabecular structures running buccolingually. (b) Areas with small virtual HU values (578) were not included in the measurement, and areas with HU values of >1000 (red arrows) were measured to obtain the coronal width (CW). (**B**) Sagittal section of the same region. The sagittal width (SW) was measured along with the CW, and the mean value of the CW and SW was defined as the mandibular cortical width.

**Table 1 reports-06-00048-t001:** Correlation between MCW and patient data.

	91 Sites (70 Patients)	Correlation with MCW	
		Significant Difference	*r*
Age, years ^a^	44.1 ± 14.9	ns	−0.116
	44 (20–80)		
Present teeth ^b^	26.8 ± 4.5	ns	0.151
	28 (4–32)		
MCW, mm ^c^			
MCI class 1 (53 sites)	3.52 ± 0.71	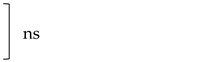	
	3.41 (2.49–6.05)		
MCI class 2 (35 sites)	3.44 ± 0.82		
	3.42 (1.81–5.14)		
MCI class 3 (3 sites)	2.22 ± 0.15		
	2.00 (2.0–2.3)		

Data are presented as mean ± standard deviation or median (range). ^a^ Pearson’s correlation coefficient, ^b^ Spearman’s rank correlation coefficient, ^c^ Student’s *t*-test. ns, not significant; MCW, mandibular cortical width; MCI, mandibular cortical index.

## Data Availability

Raw data were generated at Nihon University School of Dentistry, Mishima Dental Center. Derived data supporting the findings of this study are available from the corresponding author K.S. on request.

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
