# Peer review of "Diagnostic Study of Mandibular Cortical Index Classification Using Dental Cone-Beam Computed Tomography Findings: A Preliminary Cross-Sectional Study"

_reports, 2023, doi:10.3390/reports6040048_

Round 1

Reviewer 1 Report

I would like to take the present time to congratulate the authors for the work conducted in order to prepare the present manuscript.

I would like to present a few recommendations and concerns:

The Abstract does not require subheading according to the authors instructions

I recommend the authors to place the keywords by alphabetic order

The following sentence should be a part of the Discussion and not from the Introduction: “We hope that the imaging findings obtained during dental care will serve as a beter diagnostic aid in detecting hidden osteoporosis. To achieve these objectives, this single-center cross-sectional study was conducted as a descriptive epidemiological investigation.”

The aim sentence in the end of the Introduction section should be improved. The authors should be more specific regarding which points they really intent to investigate, assess and determine. What are the outcomes to assess?

How were the patients selected? All? Random? Consecutive cases? What was the sampling method?

Was this a convenience sampling?

What was the patient ethnic group?

What were the settings and resolutions of the DPR and CBCT images?

How many cases were excluded and why? What was the exclusion ratio?

May the authors describe the step-by-step assessment method?

Cohen Kappa only allows to compare two results (eventually 2 observers results), but not three. How was the inter-rater reliability assessment conducted when you have 3 observers and Cohen Kappa is only amble to compares 2?

The results section looks fine.

The Discussion looks good, I just would suggest to debate the study strength and generalization of the results.

Author Response

Thank you very much for your valuable and constructive comments. Your remarks are well-founded and gave us a better understanding how our manuscript can be improved.

Please find changes to the manuscript highlighted in yellow.

Reviewer: 1

・The Abstract does not require subheading according to the authors instructions.

Response: AGREE

Dear Reviewer, thank you very much for your detailed review. We removed the subheadings from Abstract section.

・I recommend the authors to place the keywords by alphabetic order.

Response: AGREE

We thank the reviewer for pointing this out. We placed the keywords by alphabetic order.

・The following sentence should be a part of the Discussion and not from the Introduction: “We hope that the imaging findings obtained during dental care will serve as a beter diagnostic aid in detecting hidden osteoporosis. To achieve these objectives, this single-center cross-sectional study was conducted as a descriptive epidemiological investigation.”

Response: AGREE

Dear Reviewer, thank you very much for your constructive comments. The sentences have been moved to the Discussion section.

Changes made: [Page 12, Line 4-7 in Word file]

・The aim sentence in the end of the Introduction section should be improved. The authors should be more specific regarding which points they really intent to investigate, assess and determine. What are the outcomes to assess?

Response: AGREE

Dear Reviewer, thank you very much for your kindly comment. The end of Introduction section has been revised as follows: Therefore, the purpose of this preliminary study was to numerically support for DPR image-based MCI classification and to simplify difficult diagnoses through measurement of the mandibular inferior cortical bone width in three dimensions using cone-beam computed tomography (CBCT) images. Furthermore, the ultimate goal is the imaging findings obtained during dental care will serve as a better diagnostic aid in detecting hidden osteoporosis.

・How were the patients selected? All? Random? Consecutive cases? What was the sampling method?

・Was this a convenience sampling?

Response: AGREE

Dear Reviewer, thank you very much for your detailed review. This cross-sectional study was conducted on a complete survey. The subjects were all patients who came to the hospital. For this reason, we did not do random sampling. We have revised this supplement as “All women” in the Study population section. Additionally, this is mentioned at the beginning of the results.

Changes made: [Page 6, Line 7 in Word file]

・What was the patient ethnic group?

Response: AGREE

We thank the reviewer for pointing this out. All final patients who met the criteria were Japanese. This description has been added to the results.

Changes made: [Page 11, Line 6 in Word file]

・What were the settings and resolutions of the DPR and CBCT images?

Response: AGREE

We thank the reviewer for pointing this out. Image data capture section has been revised as follows: All DPR images and CBCT images used in this study were taken with a ProMax 3D X-ray machine (Planmeca, Helsinki, Finland) in the clinic. The imaging conditions were as follows: DPR images, tube voltage 66 kV, tube current 9 mA, and irradiation time 19 s, CBCT images, voxel size 200 μm × 200 μm × 200 μm, tube voltage 90 kV, tube current 10 mA, and irradiation time 12 s. The diagnostic imaging software used was the Romexis 2D and 3D imaging module (Planmeca).

Changes made: [Page 7, Line 1-6 in Word file]

・How many cases were excluded and why? What was the exclusion ratio?

Response: AGREE

Dear Reviewer, thank you very much for your kindly comment. As stated at the beginning of the Results section, only one patient was excluded, who had a segmental mandibulectomy. One patient out of 71 was excluded, resulting in an exclusion ratio of 1.4 %. This has been added to the Results section.

Changes made: [Page 11, Line 5 in Word file]

・May the authors describe the step-by-step assessment method?

Response: AGREE

Dear Reviewer, thank you very much for your constructive comments. Following the reviewer's advice, an outline of the steps of the study process was added (the end of Image data capture section)

Changes made: [Page 7, Line 16-18 in Word file]

・Cohen Kappa only allows to compare two results (eventually 2 observers results), but not three. How was the inter-rater reliability assessment conducted when you have 3 observers and Cohen Kappa is only amble to compares 2?

Response: AGREE

Dear Reviewer, thank you very much for your constructive comments. We are sorry for the lack of explanation. As the reviewer stated, Cohen Kappa should be considered by two examiners. Although there were three examiners in this study, interindividual reproducibility was examined three times (dentist 1 and 2, dentist 2 and 3, dentist 1 and 3). This gives three results, which are already listed in the Results section.

Changes made: [Page 8, Line 9-12, Page 11, Line 11-13 in Word file]

・The results section looks fine.

Dear Reviewer, thank you very much for your kindly comment.

・The Discussion looks good, I just would suggest to debate the study strength and generalization of the results.

Response: AGREE

We thank the reviewer for pointing this out. The following sentence was added before the statement of limitation: The strength of this study is that it will contribute to presenting evidence in an unknown area that links the fields of osteoporosis treatment and dentistry.

In addition, we added the following generalized sentence at the end of the conclusion:

In the future, dentists treating the older people should be aware of these characteristics, which may contribute to reducing the severity of osteoporosis.

Changes made: [Page 14, Line 9-10, Page 15 in Word file]

Reviewer 2 Report

This brief report is well written, however, the authors’ need to check their results in table 1, I cannot find any results for “*p < 0.05, **p < 0.01” in the table.

Author Response

Thank you very much for your valuable and constructive comments. Your remarks are well-founded and gave us a better understanding how our manuscript can be improved.

Reviewer: 2

・This brief report is well written, however, the authors’ need to check their results in table 1, I cannot find any results for “*p < 0.05, **p < 0.01” in the table.

Response: AGREE

Dear Reviewer, thank you very much for your detailed review. We removed the p-values from Table 1.

Reviewer 3 Report

Dear Authors,

I’ve extensively read the manuscript titled “Diagnostic study of mandibular cortical index classification using dental cone-beam computed tomography findings: A preliminary cross-sectional study”. The aim of this preliminary cross-sectional study is the analysis of the cone-beam computed tomography (CBCT) data using the results to examine an objective classification of MCI.

Some aspects must be addressed before considering the manuscript suitable for publication:

-       A moderate revision of scientific English language is required.

-       Please provide the institution e-mail addresses for all the authors, as foreseen by the template

-       The abstract needs to be re-structured by MDPI guidelines. Add a clear purpose for your work.

-       In the introduction and abstract make the aim of the study clearer.

-       In the introduction you should argue the measurement of the cortical starting from CBCT and the new technologies. I suggest reading and adding: PMID: 36496410, PMID: 34553817, PMID: 32575875.

-       "The target area was immediately below the mental foramen". make it clearer which section is taken in the CBCT.

-       What are the future prospects?

-       The conclusion section needs to be revised with a more clear outcome of the study.

To sum up, article should be reconsidered after major revision

Moderate editing of English language required

Author Response

Reviewer: 3

Dear Authors,

I’ve extensively read the manuscript titled “Diagnostic study of mandibular cortical index classification using dental cone-beam computed tomography findings: A preliminary cross-sectional study”. The aim of this preliminary cross-sectional study is the analysis of the cone-beam computed tomography (CBCT) data using the results to examine an objective classification of MCI.

Some aspects must be addressed before considering the manuscript suitable for publication:

Thank you very much for your valuable and constructive comments. Your remarks are well-founded and gave us a better understanding how our manuscript can be improved.

Please find changes to the manuscript highlighted in yellow.

・A moderate revision of scientific English language is required.

Response: AGREE

Dear Reviewer, thank you very much for your constructive comments. We used an English editing service and added the following sentence in the acknowledge: The authors also thank Angela Morben, DVM, ELS, from Edanz (https://jp.edanz.com/ac) for editing a draft of this manuscript.

Changes made: [The end of page 16 in Word file]

・Please provide the institution e-mail addresses for all the authors, as foreseen by the template

Response: AGREE

We thank the reviewer for pointing this out. We will provide them in the template at the time of submission.

・The abstract needs to be re-structured by MDPI guidelines. Add a clear purpose for your work.

・In the introduction and abstract make the aim of the study clearer.

Response: AGREE

Dear Reviewer, thank you very much for your kindly comment. First, we removed the subheadings from Abstract section. Next, we added clear objectives to the Abstract and Introduction sections.

・In the introduction you should argue the measurement of the cortical starting from CBCT and the new technologies. I suggest reading and adding: PMID: 36496410, PMID: 34553817, PMID: 32575875.

Response: AGREE

Thank you very much for your valuable and constructive comments. The following sentence was added in the Introduction section and subsequent reference numbers were corrected: The accuracy of 3D reconstructions of the craniomaxillofacial region using cone beam computed tomography (CBCT) is important for the morphological evaluation of mandibular cortical bone anatomical structures [16-18].

Changes made: [Page 5, Line 6-8 in Word file]

・"The target area was immediately below the mental foramen". make it clearer which section is taken in the CBCT.

Response: AGREE

Dear Reviewer, thank you very much for your detailed review. The following description has been added: (area indicated as red arrow in Fig 2, B)

Changes made: [Page 9, Line 2-3 in Word file]

・What are the future prospects?

・The conclusion section needs to be revised with a more clear outcome of the study.

Response: AGREE

Thank you very much for your valuable and constructive comments. We added the following generalized sentence at the end of the conclusion: In the future, dentists treating the older people should be aware of these characteristics, which may contribute to reducing the severity of osteoporosis.

Changes made: [Page 15 in Word file]

・To sum up, article should be reconsidered after major revision

Finally, we would like to thank you for your valuable efforts to improve the quality of this manuscript. We believe that your comment significantly improved the manuscript and will help readers interpret and implement the knowledge transferred in the present manuscript.

Round 2

Reviewer 1 Report

Dear author, I have no more concerns.

Reviewer 2 Report

No more comments

Reviewer 3 Report

I am delighted to inform you that your revisions have been thoroughly reviewed, and I am pleased to say that the article is now ready for publication.